# Analysis and Design of a Diplexing Power Divider for Ku-Band Satellite Applications

**DOI:** 10.3390/s23218726

**Published:** 2023-10-26

**Authors:** Farzad Karami, Halim Boutayeb, Ali Amn-e-Elahi, Larbi Talbi, Alireza Ghayekhloo

**Affiliations:** Department of Computer Science and Engineering, University of Quebec in Outaouais (UQO), Gatineau, QC J8Y 3G5, Canada; karf06@uqo.ca (F.K.); larbi.talbi@uqo.ca (L.T.); alireza.ghayekhloo@uqo.ca (A.G.)

**Keywords:** diplexing device, bandpass filters, Ku-band systems, space probe applications, power dividers

## Abstract

In dual-band RF front-end systems, to transmit different frequency signals in different paths, each path requires the power to be divided along two transmission channels. In such systems, a circuit is created in which the input ports of power dividers with different frequency bands are connected to the output ports of a diplexing circuit in a cascade form. These circuits often contain different band filters in their schemes and have a complicated design. In this paper, an innovative technique for designing a diplexing power divider for Ku-band applications is presented. The proposed structure is designed on multilayer printed circuit boards (PCBs) and the utilization of a transition based on an extended SMA connector. The extended SMA connector provides two separate paths for the transmission of the RF signals. Hence, the proposed structure eliminates the need for intricate and bulky bandpass filters, allowing seamless integration with other planar devices and components within Ku-band satellite subsystems. In fact, the proposed architecture channelizes the divided output electromagnetic signals into two separate frequency spectrums. With the presented technique, two frequency ranges are envisaged, covering Ku-band applications at 13–15.8 GHz and 16.6–18.2 GHz. With the proposed structure, an insertion loss as low as 1.5 dB was achieved. A prototype of the proposed power-divider diplexing device was fabricated and measured. It exhibits a good performance in terms of return loss, isolation, and insertion losses.

## 1. Introduction

Recent years have seen the development of highly integrated multifunction devices as an effective means of achieving high-performance and low-cost solutions. Incorporating solar panels into electronic devices is a prime example of multifunctionality [1,2]. This approach is particularly crucial in space communications where sunlight energy harvesting is required [3,4]. Power dividers and diplexers are pivotal elements in RF systems and instruments. They find extensive application in novel space telecommunications for segregating dual communication pathways. In wireless systems, power dividers are commonly used for power division and power combining with proper isolation and frequency filtering features [5,6,7,8,9]. Diplexing circuits play a significant role in transceiver front-end wireless systems. They can effectively separate Tx/Rx channels connected to common wideband or dual-band antennas [10,11,12,13]. Therefore, a diplexer technique can be employed to accommodate a system with multiple frequency bands and multiple functionalities. They can be used, for example, in a deep-space probe antenna system, in which the transmit and receive signals are divided to cover a wide communication delay and the space probes are assumed to be far from Earth [14]. Diplexing circuits provide essential filtering characteristics and isolation between Tx and Rx channels in multi-input and multi-output services for a space link. In this scenario, diverse communication links are harnessed to transmit or receive signals from the deep-space probe. This covers extensive distances while minimizing potential interference [15,16].

Diplexers are often designed using two bandpass filters with coupled resonators or cavities [17,18,19,20]. However, using a combination of low-pass and high-pass filters rather than two bandpass filters is preferable in some cases where the relative bandwidth of the transmit and receive channels is large [21]. Conventional diplexers are usually designed by using a rectangular waveguide, metal cavity junctions, or gap waveguide topologies to achieve desirable filtering performance [22]. However, when implemented in a practical array of probe antennas, these configurations can be cumbersome, heavy, and financially challenging [23], particularly in the context of space explorations in the future.

One of the main challenges for modern circuit designers is to design high performance, modular, and integrated microwave components with a compact topology [24,25,26]. Several researchers have been focused on designing planar diplexing systems [27,28]. Certain types among them necessitate the utilization of the low-temperature cofired ceramic (LTCC) fabrication process to attain a more compact (packed) and densely arranged (miniaturized) device. Other types of these components employ microstrips, striplines, planar traces, and coplanar waveguides (CPW). Typically, these components comprise a duo of bandpass filters functioning in distinct frequency bands, interconnected through a distribution network [29,30,31]. These combined distribution networks often occupy considerable space and lead to an increase in the overall size of the circuits. Although these planar architectures present good performance at lower frequencies, they strongly suffer from undesired emission and radiation at high frequencies.

Substrate integrated waveguide (SIW) is a renowned and currently well-established technological planar platform for RF, microwave, and millimeter-wave communication systems [32,33,34]. SIW is the most effective alternative to traditional microstrip or coplanar waveguides at high frequencies with extra performance. They have many features of rectangular waveguides, with the advantage of a low-cost fabrication process based on regular printed circuit board (PCB) technology [35]. SIW technology has found application in the design of various microwave and millimeter-wave components, such as filters, sensors, and reconfigurable electronics [36,37,38]. This technology offers compact dimensions, cost-effectiveness, high quality, and exceptional performance. To develop SIW duplexing devices (which means the separation of the functions of transmission and reception), a T-junction is often required to provide the matching and isolation requirements between the input port and two separate channel filters. This T-junction occupies a certain space in architecture [39]. Dual-mode duplexing can remove T-junction size overhead while maintaining design flexibility and performance improvement. These structures have a compact size, but suffer from a narrow operating bandwidth.

Designing high-performance and compact components with integration and modularity features for radio frequency front ends is highly desirable in many communication systems [40]. The concept of modularity presents a promising avenue for seamlessly integrating cutting-edge electronics, an apt approach to the cost-effective implementation of fully tailored and adaptable electronic devices. By embracing modularity, the capacity to assemble and integrate individual modules with distinct functionalities provides a flexible framework for crafting customized and responsive electronic systems that align precisely with specific requirements. This approach not only streamlines the development process but can also significantly reduce the costs associated with intricate electronic device realization [41,42]. A lot of interest has been attracted to the integration of multifunctional microwave components, such as multiband and filtering dividers [43,44,45,46], diplexer antennas [47,48,49,50], etc. In long-distance communication systems, low-profile diplexing devices are paramount. These devices play a pivotal role in efficiently guiding and separating electromagnetic signals along designated pathways. By integrating these compact and versatile diplexers into the communication infrastructure, the transmission and reception of signals across long distances are optimized. This optimization is achieved by strategically channeling distinct frequency bands through the diplexing components. This enhances the overall efficacy and reliability of the communication system. The utilization of these diplexing devices improves signal management and reduces interference and signal degradation. This ensures seamless and dependable long-distance communication. The integration of these components with power dividers could benefit from size reduction and loss reduction in antennas used, for example, in deep space telecommunications.

This paper presents the analysis and design of an innovative compact Ku-band SIW duplexing power divider. The proposed concept is based on extended SMA connectors, which are used as a transition between two stacked SIW channels. This innovative approach aims to achieve improved signal integrity and efficient signal propagation in Ku- band frequency applications. The proposed transition behaves like two pairs of bandpass filters effectively separating the channels. However, the unique aspect lies in its compact size, which the occupied size is much smaller than the size of two regular bandpass filters. The developed duplexing power divider has a simple topology and shows great potential for excellent integration with other planar components. The remainder of the paper is organized as follows. An SIW diplexing circuit and its characteristics are described in Section 2. In this section, a parametric study is provided to show the effect of some important parameters on S-parameters. In Section 3, the procedure for designing a diplexing power divider is presented. Then, its experimental results are presented and discussed in this section. Finally, concluding remarks are given in Section 4.

## 2. Geometry of the Proposed Diplexing Structure and Performance Analysis

### 2.1. Overall Structure

The geometry of the proposed SIW diplexing network is depicted in Figure 1, encompassing all aspects of design, layers, and the advanced SMA port. It is composed of two stacked SIW channels from top to bottom, including two Ro 4003c PCB laminates (*ε*r = 3.55 and tanδ = 0.0027). Both PCB laminates have the same height of 1.6 mm.

There is a whole copper layer between the two PCB laminates, the thickness of which is 4.5 mm. Figure 1a shows the complete geometry of the proposed structure. In Figure 1b, the copper layer has been hidden for a better view of the internal connections in the proposed structure. The top view of PCB laminates and the copper layer is shown in Figure 1e.

The proposed structure has three ports, which are numbered 1, 2, and 3. Ports 1 and 2 are made with two 50-ohm typical SMA connectors, as shown in Figure 1b with blue arrows. Port 3 is implemented using an SMA connector of 50 Ohms with an extended dielectric. The side view of the proposed structure is shown in Figure 1c.

The end pin of the SMA connectors 1 and 2 are soldered in the center of the SIW channel axis. Connector 3 has an extended dielectric, as shown in Figure 1d. The extended dielectric SMA port is the prepared interface that seamlessly penetrates the central copper layer without interference. The material of this dielectric is PTFE (*ε*r = 2.1 and tanδ = 0.0002). The PTFE dielectric is 4.5 mm long. It can be easily removed from the connector metallic pin for structure assembly. The cylindrical dielectric has an inside diameter of 4.1 mm, and an outside diameter of 1.27 mm. Figure 1d illustrates the extended dielectric and its connector together.

The proposed diplexing network is made up of three stacked PCB layers, as can be seen in Figure 1. The primary component of the suggested architecture is the vertical transition. The transition starts at the top layer, and its end is soldered to the bottom copper layer. In this structure, the cut cylindrical PTFE (shown in Figure 1d) is placed inside a hole made in the copper layer. In the copper layer, there is a hole whose diameter is 4.1 mm (the same as the outer diameter of PTFE). The copper layer serves as a protective electromagnetic shield for the PTFE-covered conductor. The termination point of SMA connector pin 3 involves soldering it to the lower ground of the PCB’s bottom layer, following its traversal through the first PCB layer, the copper layer, and eventually the lower PCB layer. This meticulous arrangement ensures optimal signal integrity and grounding in the design. In contrast to connectors 1 and 2, connector 3 is not located on the central axis of the SIW channel. The distance between SMA connector 3 and the SIW center axis is 3.2 mm.

### 2.2. Analysis

In the proposed design, signals between SIW channels are then divided or separated by shifting the vertical transition from the central axis to the narrow walls of the SIWs. Dividing or separating electromagnetic signals between SIW channels depends on the position of SMA connector 3. By relocating SMA connector 3 from the center of the SIW axis to its walls, the suggested topology modification has the potential to alter transition behavior significantly. Illustrated in Figure 2 are the scattering parameters of SMA connector 3, demonstrating an extended dielectric connector’s role as via a coupling topology when situated along the central axis of the SIW. In this particular arrangement, the RF signal undergoes a division between SIW channels 1 and 2. Assigned as the input port, port 3 serves as the point of entry for the RF signal, which subsequently becomes distributed between the two aforementioned SIW channels (channels 1 and 2). This configuration showcases how the placement of SMA connector 3 along the SIW axis center can facilitate signal division and coupling through the extended dielectric connector. It is important to note that this arrangement holds implications for signal distribution and interaction within the SIW structure. Engineers and designers can leverage this insight to optimize signal routing, distribution, and interaction within the SIW, thereby tailoring the topology to specific performance objectives. The coupling values between the two output ports exhibited a notable similarity, while the reflection coefficient of port 3 was sufficiently low.

By moving the transition toward the SIW walls, a distinct separation of electromagnetic signals emerges, channeling them into channels 1 and 2 at varying frequency bands. For optimization, the parameters Hs, Xoffset, Y_1_, and Y_2_ are changed. These four parameters have the most effect on scattering parameters. By working on these parameters, shifts occur in the values of inductance, capacitive reactance, and conductance within the impedance characteristics of the system. To determine the optimal values for these key parameters, comprehensive full-wave simulations employing CST software were executed. An evaluative exploration is undertaken through a parametrical analysis, depicted in Figure 3, showing the impact of these parameters. Notably, only a singular parameter is altered in each analysis iteration, maintaining constancy across all other parameters. Between these parameters, it seems that Hs, Y_1_, and Y_2_ cause S33, S31, and S32, respectively, to shift.

Figure 4 illustrates the distribution of the electric fields on the SIW channels at 14 GHz (low-frequency band) and 17 GHz (high-frequency band) of the diplexing network. In this case, the mechanism’s topology can be comprehended. If port 1 is excited in the low-frequency band, the RF signal first propagates in SIW channel 1. After that, the propagated signal is vertically transitioned by the proposed transition. Once the signal is propagated into the vertical transition, port 3 becomes active. In this configuration, no RF signal propagates into SIW channel 2. Channels 1 and 2 are thus isolated from one another. If port 2 is excited in the high-frequency band, the RF signal is transmitted in SIW channel 2, and it enters the proposed vertical transition at the end of channel 2. After that, it reaches the vertical transition, and port 3 becomes active. In this configuration, no RF signals pass through SIW channel 1. Channel 1 and channel 2 are, therefore, again isolated.

Likewise, by exciting port 3 with a wide-frequency band signal, signals within SIW channels 1 and 2 are propagated across distinct frequency bands. The vertical transition channelizes the RF signal between SIW channels 1 and 2 if port 3 is excited.

The simulated scattering parameters (S-parameters) of the optimized diplexing network are graphically depicted in Figure 5. Within this representation, it becomes evident that the lower channel (SIW channel 1) demonstrates an insertion loss of less than −1.5 dB, spanning the frequency range of 13 to 15.8 GHz, which corresponds to approximately 19.45% of the total bandwidth (Figure 5a). Concurrently, in the frequency interval of 16.6 GHz to 18.2 GHz, encompassing about 9.3% of the bandwidth, the upper channel (SIW channel 2) exhibits an insertion loss also below −1.5 dB (Figure 5b).

Further analysis of the simulation reveals matching bandwidths for amplitudes of S_11_ and S_22_ that are less than −10 dB. Specifically, the optimized diplexing network presents a matching bandwidth of around 20%, spanning from 12.6 to 15.4 GHz, for S_11_. For S_22_, the corresponding matching bandwidth covers a range of approximately 9.15%, spanning from 16.7 to 18.3 GHz. These findings underscore the effectiveness of the optimized network in achieving favorable insertion losses and suitable matching bandwidths, thereby substantiating its potential for efficient diplexing operations.

## 3. Diplexing-Power Divider: Design and Performance

The geometry of the proposed diplexing structure demonstrates its seamless integration potential with various planar microwave components, serving as a versatile modular device. This integration capability positions it favorably for inclusion in communication systems such as space applications when the necessitating dual operating frequency bands are met. The configuration of the ultimate diplexing power divider is depicted in Figure 6. This arrangement involves the interconnection of two pairs of Y-junction power dividers with the output ports of the aforementioned diplexing structure. In Figure 6a, the output ports for each power divider are denoted as SMA connectors 3 and 4. Notably, the structure geometry was optimized using the CST software suite.

The simulated S-parameters of the proposed diplexing-power divider are presented in Figure 7, originating from the simulation software. They show that the amplitude of the transmission coefficients from port 1 to port 3 or 4 surpasses the −4.5 dB threshold (Figure 7a). The same result is obtained when observing the transition from port 2 to ports 3 and 4 (Figure 7b). Given the inherent necessity to distribute power across two distinct ports, we can conclude that the achieved insertion loss was less than 1.5 dB. The impedance bandwidths for amplitudes of S_11_ and S_22_ less than −10 dB is 10.2% (13 GHz to 14.4 GHz) and 9.2% (16.6 GHz to 18.1 GHz), respectively.

Isolation between uncoupled ports (ports 1 and 2) is more than 17 dB in the low-frequency band and more than 15 dB in the high-frequency band, based on the achieved charts in Figure 7. If port 1 is excited in the low-frequency band, the RF signal propagates into the lower power divider. After that, the propagated signal is vertically transitioned into the vertical transition. Once the signal is propagated into the vertical transition, ports 3 and 4 become active. The amplitude and phase of the signal at ports 3 and 4 are equal. In this situation, ports 1 and 2 are isolated from one another. On the other hand, if port 2 is excited in the high-frequency band, the RF signal is transmitted in the top power divider and it enters the vertical transition at the end of the top power divider. After it reaches the vertical transition, ports 3 and 4 become active. In this configuration, channel 1 and channel 2 are again isolated.

For further assessment of the operational dynamics of the final power divider employing this compact module, an analysis of electric fields and their trajectories is prepared in Figure 8. In particular, Figure 8a portrays the scenario wherein port 1 serves as the designated input port, effectively isolating it from any potential interference between port 2. This configuration illuminates the transmission path of the wave reaching the output ports with equitably distributed power. Similarly, Figure 8b captures another electric field distribution path. This time port 2 is the input source and equal power is divided between the two outputs and no coupling to the other input connector is observed.

As the last step towards validating the simulated performance of the proposed diplexer-power divider, a tangible prototype was fabricated and tested, as shown in Figure 9. During the assembly process, the layers were placed on top of each other and then screwed together (via threaded connections). The metal pins of the SMA connectors 3 and 4 enter the top PCB layer and pass through the copper and the second PCB layer. Then, this metal pin is soldered to the copper layer at the bottom of the second PCB layer.

Figure 10 shows the measured S-parameters of the prototype, which are in good agreement with the previously presented simulated results. These acceptable results from two different solutions underscore the reliability of the proposed design. Figure 11 shows a photograph of the measurement setup with the help of a facilitated vector network analyzer (VNA) and high-frequency cables. A short, open, match load, through (SOLT) technique with mechanical kits was employed to calibrate the system for the desired frequency to achieve precise scattering parameters. In this step, a short (a piece of coaxial cable with a short circuit at the end) was connected to the VNA and performed a short calibration. Then, an open (a piece of coaxial cable with an open circuit at the end) was connected to the VNA and performed an open calibration. Finally, a load (a piece of coaxial cable with a matched impedance at the end) was connected to the VNA and performed a load calibration. In addition to the open, short, and load standards, a through calibration has been used to improve accuracy.

A comparison between various characteristics of the proposed design and previously reported results in terms of operation frequency, fractional bandwidth, insertion losses, and return losses is provided in Table 1. Several techniques have been reported in the literature to design diplexing power dividers [51,52,53,54]. Even though these techniques result in good performance, their bandwidth is still limited. The implementation of some of these topologies requires several band-pass filters, increasing the complexity of the design.

A Ku-band frequency power divider is used as a subsystem of a satellite for Earth observation and climate monitoring. This system can be equipped with various sensors and instruments to collect data on temperature, humidity, atmospheric composition, and other climate-related parameters from space satellites. The assigned frequency bands split the incoming and outgoing signals, which is a commercial and critical feature for satellite subsystems. The split channel enables signals from the power divider onboard to be transmitted to processing units. These units process the data in real-time to extract meaningful information, such as temperature trends, cloud cover, and greenhouse gas concentrations. Therefore, the proposed Ku frequency power divider is crucial to ensuring efficient and reliable communication between the satellite and ground stations. It allows the timely transmission of critical climate data. This is essential for monitoring and remotely sensing climate change, understanding its impact, and formulating policies to address environmental challenges.

## 4. Conclusions

A new technique for integrating diplexers and power dividers was proposed and validated experimentally at Ku-frequency with the aim of applications in space communication. For space exploration and deep space probe observation, precise multiband and multifunction communication devices are required. The proposed device is based on extended connectors, via transitions, substrate integrated waveguides and stacked printed circuit boards that make it a modular device. Two different frequency spans of 13–15.8 GHz and 16.6–18.2 GHz are envisioned with this technique covering Ku-band applications. The achieved insertion loss for the power divider was less than 1.5 dB. This compact structure with diplexing power dividing features was designed, fabricated, and tested in various solutions.

## Figures and Tables

**Figure 1 sensors-23-08726-f001:**
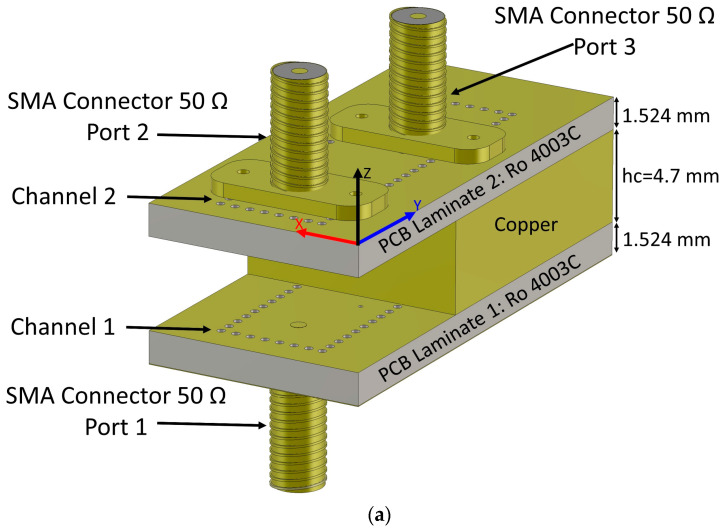
Different views of the proposed diplexing network. (**a**) Full view schematic, (**b**) without copper layer (in place material) view, (**c**) side view, (**d**) view of the advanced SMA connector with extended dielectric, and (**e**) top view of each of the layers (all dimensions are in mm).

**Figure 2 sensors-23-08726-f002:**
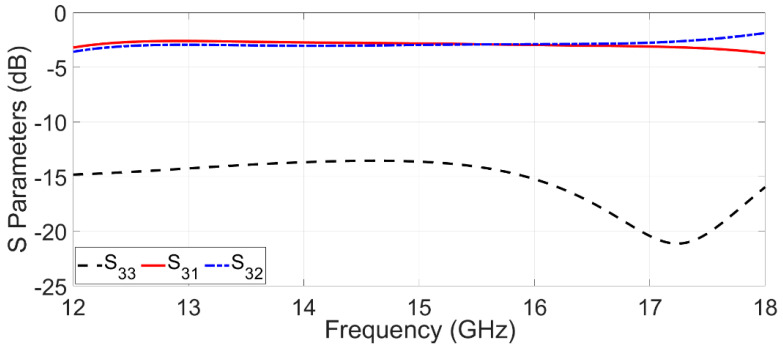
Simulated S-parameters of the proposed structure if the vertical transition is positioned at the SIW center axis.

**Figure 3 sensors-23-08726-f003:**
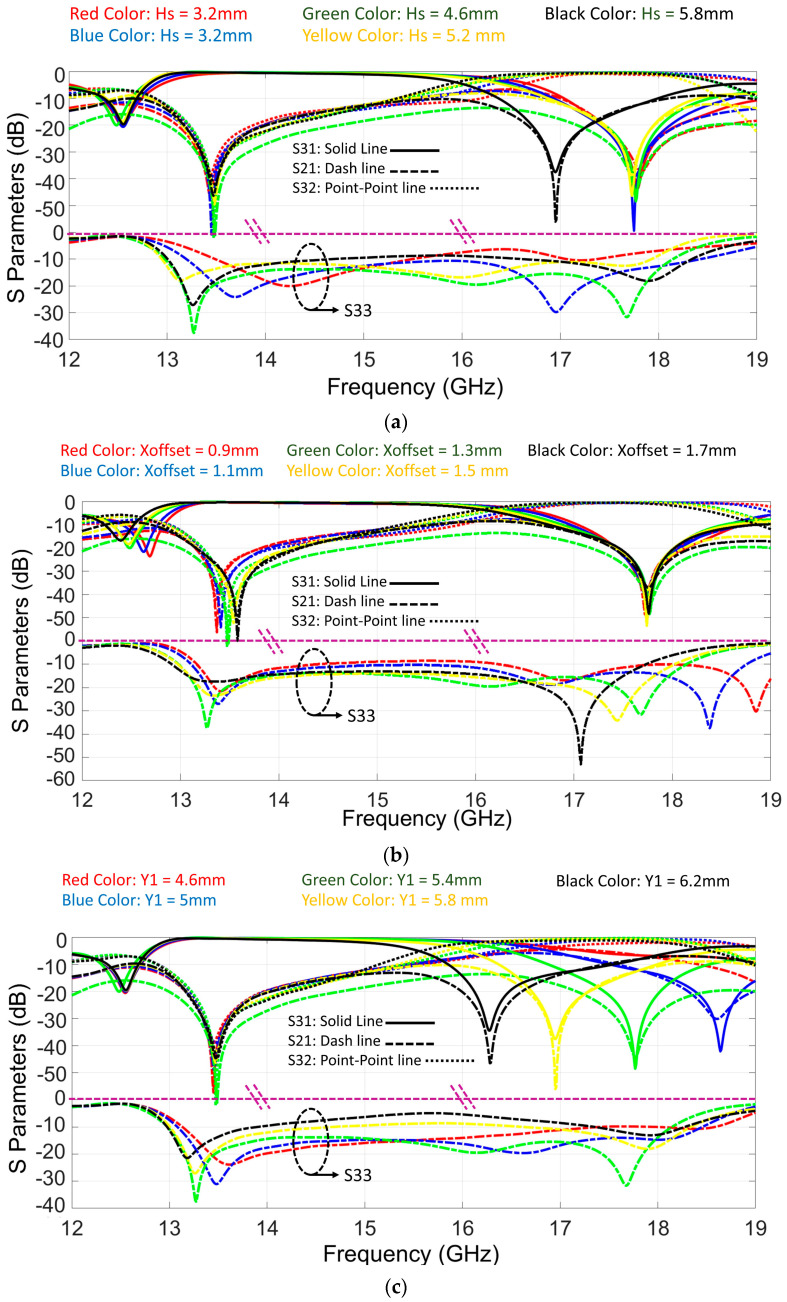
Parametric studies of the proposed structure. (**a**) Different values of thickness of copper layer, Hs from 3.2 to 5.8 mm. (**b**) Different values of Xoffset from 0.9 to 1.7 mm. (**c**) Different values of Y_1_ from 4.6 to 6.2 mm. (**d**) Different values of Y_2_ from 8.4 to 10 mm. (Solid lines, dash lines, and point-point lines represent the S31, S21, and S32 parameters, respectively, while the dash circle specifies charts for S33).

**Figure 4 sensors-23-08726-f004:**
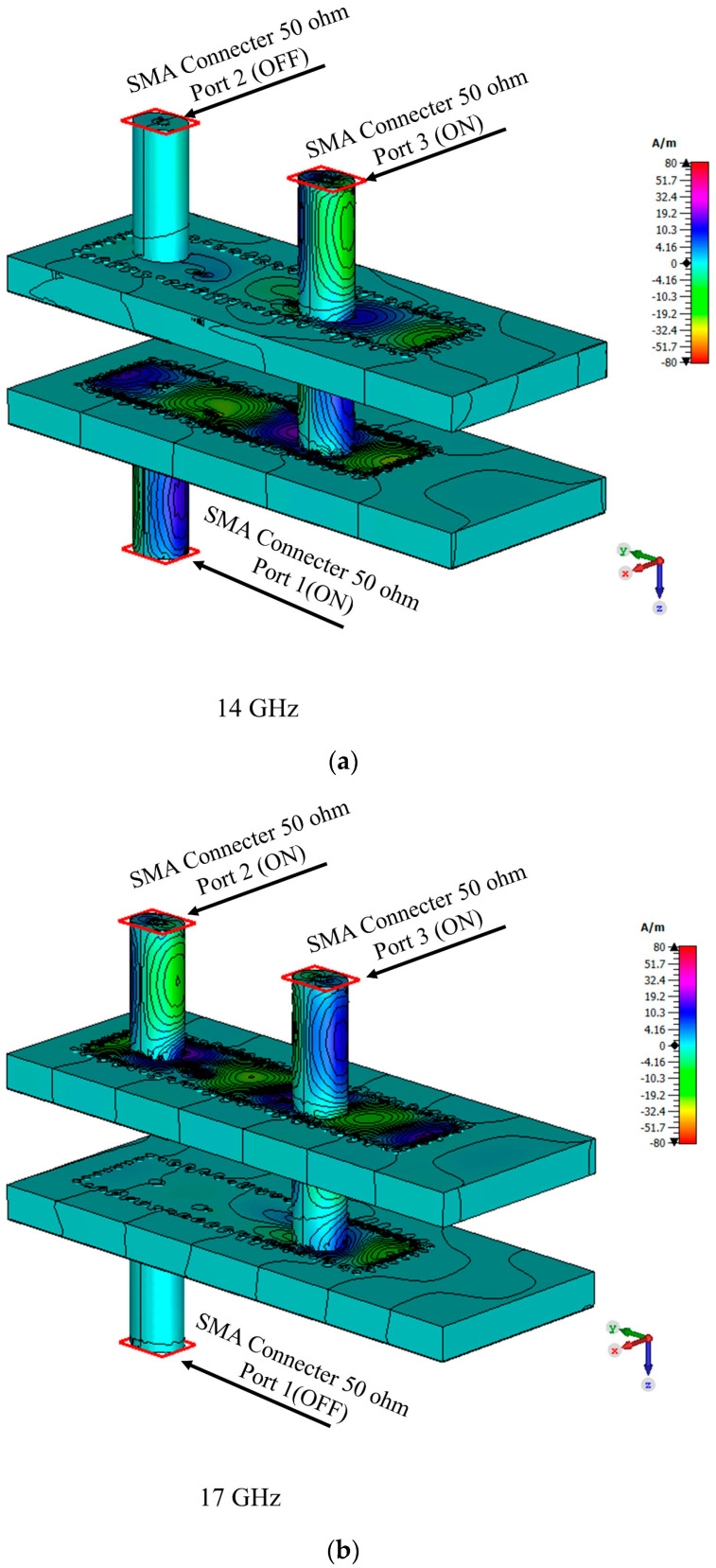
Simulated electric field distribution for the proposed architecture and two different frequencies; (**a**) when exiting port 1, and (**b**) when exiting port 2.

**Figure 5 sensors-23-08726-f005:**
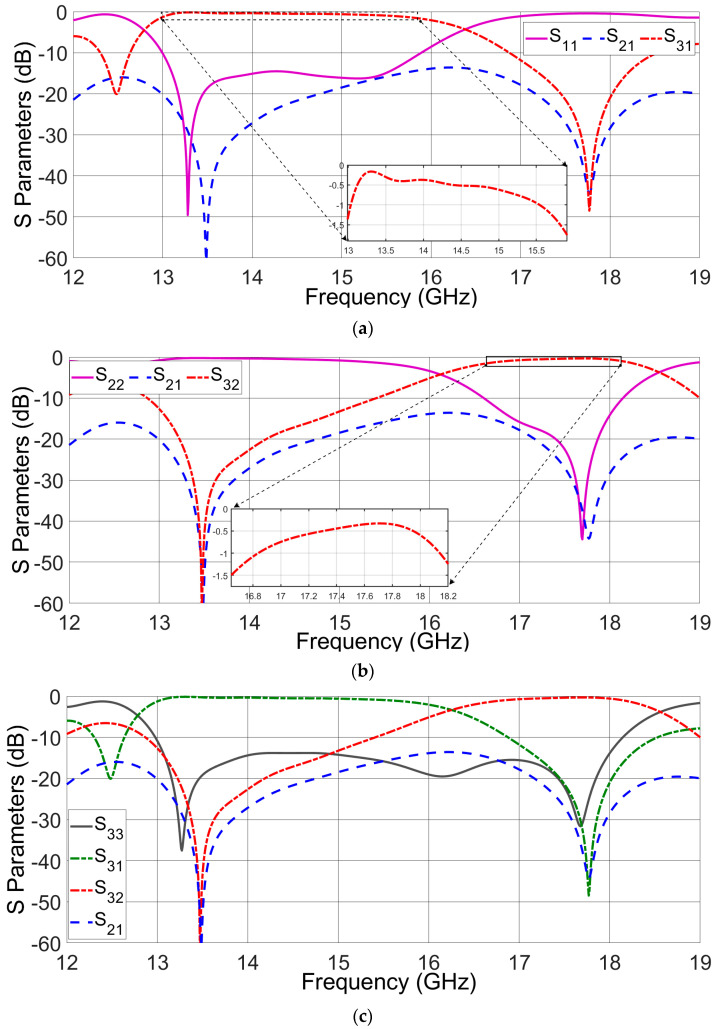
Simulated scattering parameters for the proposed diplexing structure. (**a**) Port 1 is excited. (**b**) Port 2 is excited. (**c**) Port 3 is excited.

**Figure 6 sensors-23-08726-f006:**
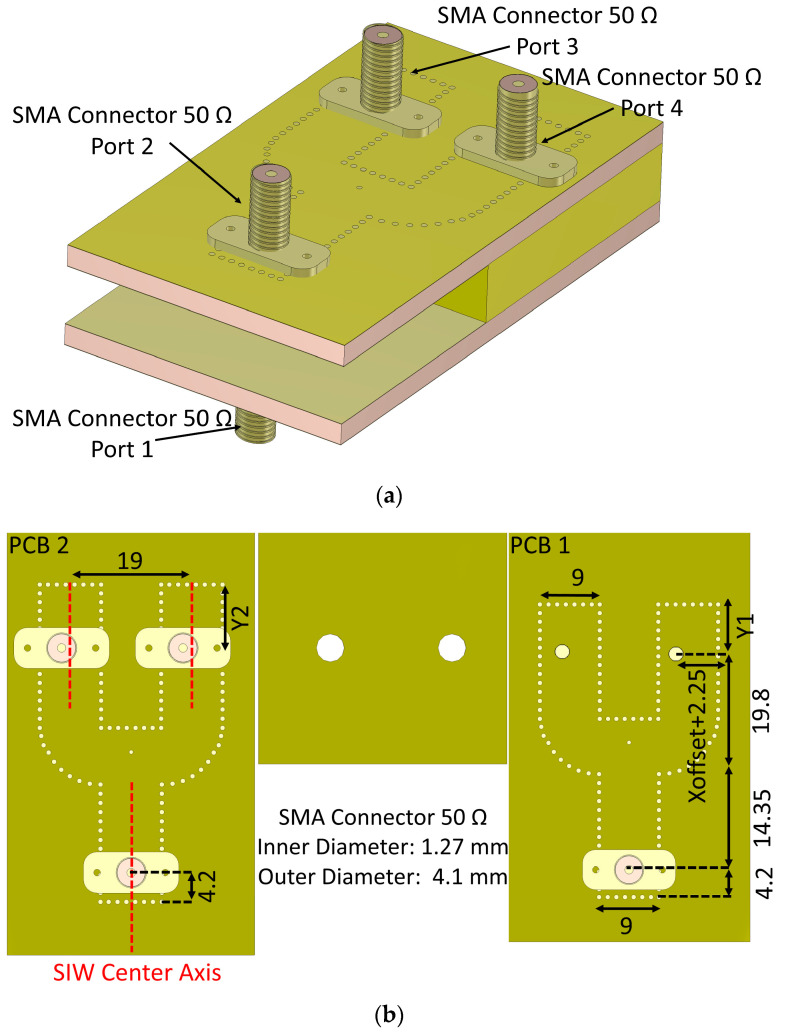
Different views of the final diplexing-power divider system working at the Ku frequency band. (**a**) Configuration of the proposed architecture, and (**b**) top views of each layer (all dimensions are in mm). (Xoffset: 1.7 mm, Y_1_: 5.9 mm, Y_2_: 10.1 mm).

**Figure 7 sensors-23-08726-f007:**
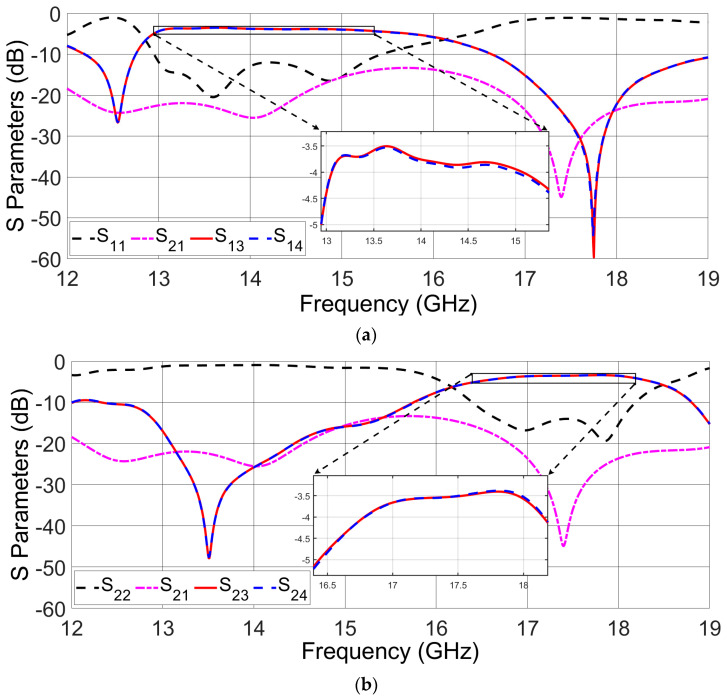
Simulated S-parameters for the final duplexing-power divider. (**a**) Excitation of port 1, and (**b**) the case of exciting port 2 as the input signal source.

**Figure 8 sensors-23-08726-f008:**
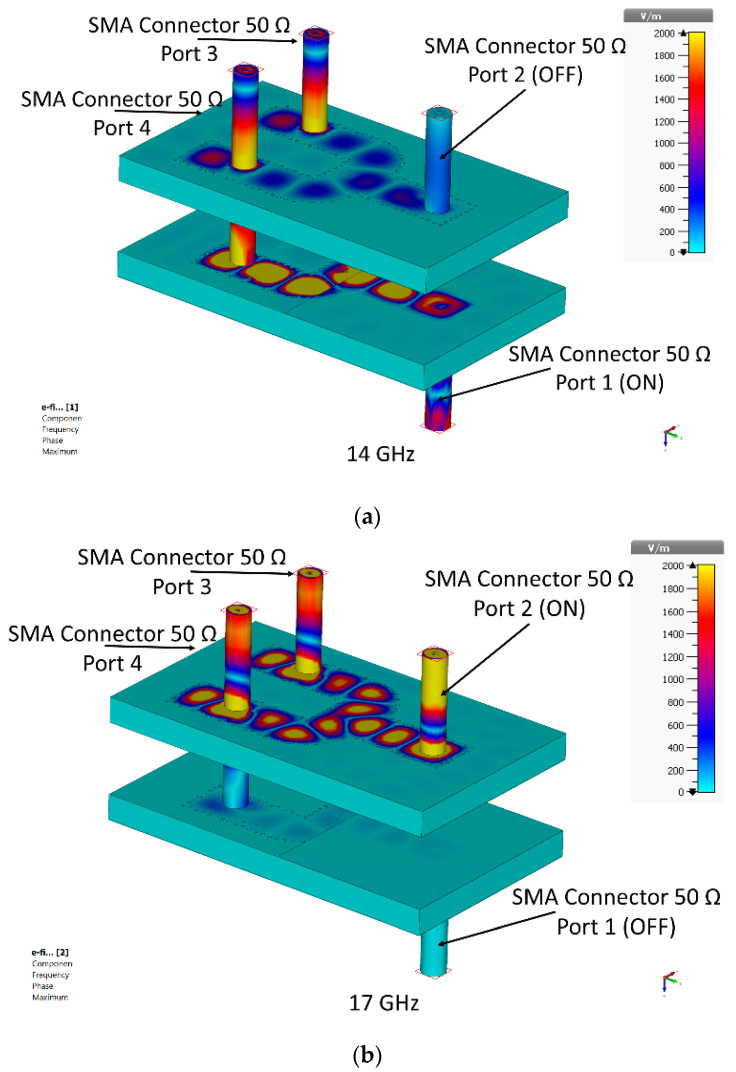
Simulated electric field distribution for proposed diplexer-power divider for different frequencies. (**a**) Exiting port 1. (**b**) Exiting Port 2.

**Figure 9 sensors-23-08726-f009:**
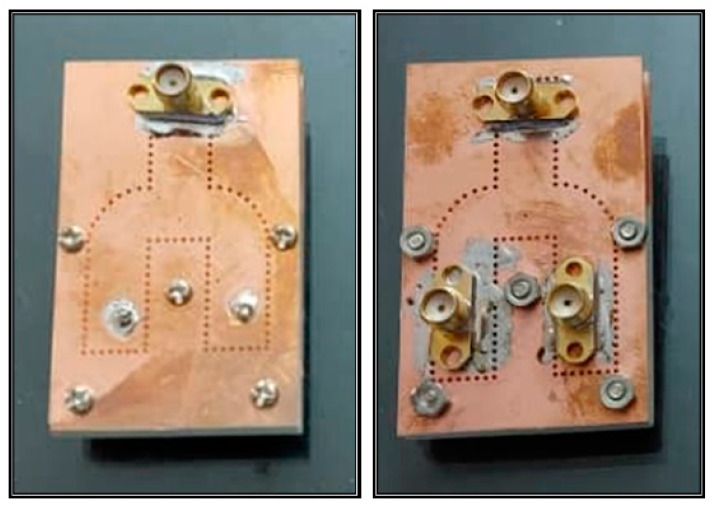
Photograph of fabricated diplexer-power divider prototype.

**Figure 10 sensors-23-08726-f010:**
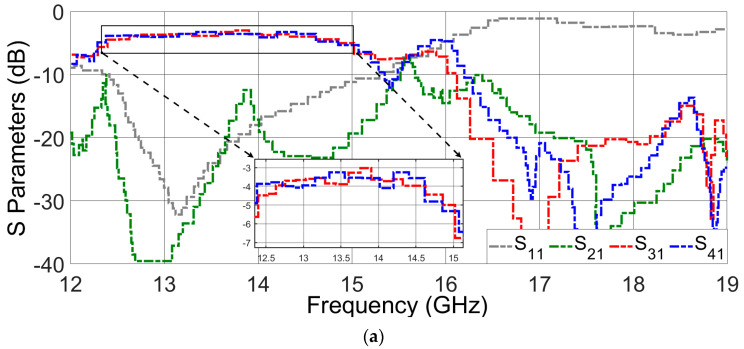
Measured S-parameters for the proposed diplexer-power divider. (**a**) Port 1 is excited. (**b**) Port 2 is excited.

**Figure 11 sensors-23-08726-f011:**
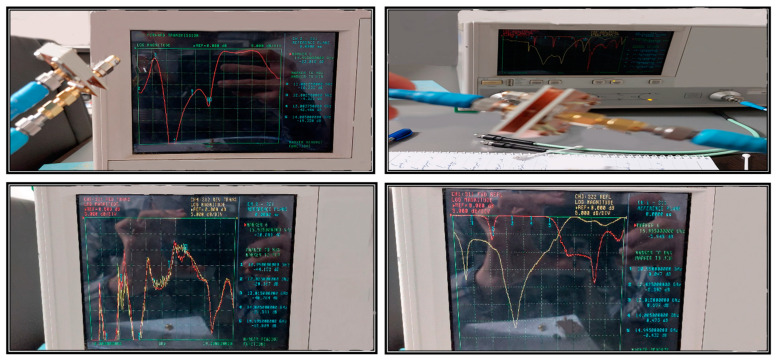
Photograph of measurement setup.

**Table 1 sensors-23-08726-t001:** A comparison of some similar works in the literature with the proposed results.

Ref.	Center Frequency (GHz)	Fractional Bandwidth (%)	Insertion Losses (dB)	Return Losses (dB)	Isolation (dB)	Structure Technology
[46]	28.2/29.2	>2.3/>2.2	0.9/0.9	13/13	55	Waveguide
[51]	73.5/83.5	>6.8/>6	Not Given	13/13	50	Waveguide
[52]	2.45/2.98	<10/<10	1.6/1.9	19.2/15.3	24	Microstrip
[53]	1.8/2.4	4.4/2.7	5.6/7	Not Given	22	Microstrip
[54]	19.8/29.7	6/4.7	1.3/1.3	10/10	25	SIW
This Work	14.4/17.4	19.4/9.2	1.5/1.5	10/10	15–40	SIW

## Data Availability

The data presented in this study are openly available.

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
