# Peer review of "Analysis and Design of a Diplexing Power Divider for Ku-Band Satellite Applications"

_sensors, 2023, doi:10.3390/s23218726_

Round 1

Reviewer 1 Report

 In this paper, an innovative technique for designing a diplexing power divider for Ku-band applications is presented. A prototype of the proposed device was fabricated and measured. It exhibits a good performance in term of return loss, isolation and insertion losses.

1-Compassion table should be added proposed designed should be compared with some related works.

2- The insertion loss and return loss values for both operating bands should be provided and compared with some related works.

3- The bandwidth and FBW parameters for both bands should be added and compared with some related works.

4-Abstract should be supported by data and results.

5- Device is fabricated and measured. The measurement setup is provided. Add more explanations about measurement setup and applied VNA.

6- Improve introduction and provide some related works in introduction section. For example, in line 47-48 “Diplexers are often designed using two bandpass filters with coupled resonators or cavities [17]-[19].” ,where BPF-BPF diplexers are described : The below diplexer can be cited.  “Design of a Compact Quad-Channel Microstrip Diplexer for L and S Band Applications. Micromachines. 2023 .”

7- Provide explanations about how authors find the dimensions in Fig.6?

8- The output ports isolation parameter (S34) is not reported at all. Provide S34 and add explanations about isolation between output ports.

Minor editing of English language required

Author Response

In this paper, an innovative technique for designing a diplexing power divider for Ku-band applications is presented. A prototype of the proposed device was fabricated and measured. It exhibits a good performance in term of return loss, isolation and insertion losses.

1- Compassion table should be added proposed designed should be compared with some related works.

2- The insertion loss and return loss values for both operating bands should be provided and compared with some related works.

3- The bandwidth and FBW parameters for both bands should be added and compared with some related works.

Your review is greatly appreciated. The revised version includes a comparison table. In red, all modifications are indicated.

4- Abstract should be supported by data and results.

In the revised version, the obtained data and results have been included in the abstract. Modifications are indicated in red.

5- Device is fabricated and measured. The measurement setup is provided. Add more explanations about measurement setup and applied VNA.

The revised version has been updated with additional explanations about measurement. The color red indicates all modifications.

6- Improve introduction and provide some related works in introduction section. For example, in line 47-48 “Diplexers are often designed using two bandpass filters with coupled resonators or cavities [17]-[19].” ,where BPF-BPF diplexers are described : The below diplexer can be cited.  “Design of a Compact Quad-Channel Microstrip Diplexer for L and S Band Applications. Micromachines. 2023 .”

In the revised version, the reference above is added.

7- Provide explanations about how authors find the dimensions in Fig.6?

In figure 6, parameters are listed in the caption. The color red indicates all modifications.

8- The output ports isolation parameter (S34) is not reported at all. Provide S34 and add explanations about isolation between output ports.

In the proposed structure, a lossless power divider (Y-Junction) is used.
Based on microwave theory, a power divider with such a structure should have an
isolation of around -6 dB between at its output ports. For the proposed design,
isolation between the output ports of the power divider (S34) is around -6 dB over
its entire operational bandwidth. The article does not mention S parameter 34
because of the large number of S parameters in the proposed structure.  

Reviewer 2 Report

The authors have chosen an interesting research topic, and have proposed a study on dilexing power divider. However, there is doubt about the novelty of the proposed study. There are some concerns;

Figure quality could be better. The coordinates of x, y and z should be drawn in Figure 1 and Figure 6.

PCB 1 and PCB 2 name tags are not compatible in Figure 1.

Is the copper layer empty box or filled block? What is the reason or effect of this copper layer being half-sized? Is it change the isolation?

My main concern is with the novelty of the design. The proposed main design topology is previously presented in [1]. In terms of performance, results could be sufficient. However, can the authors please address the uniqueness of the design in light of existing diplexing power dividers or similar design by comparing it with the previously published works? 

[1] doi: 10.1109/AP-S/USNC-URSI47032.2022.9887341.

Author Response

The authors have chosen an interesting research topic, and have proposed a study on dilexing power divider. However, there is doubt about the novelty of the proposed study. There are some concerns;

Figure quality could be better. The coordinates of x, y and z should be drawn in Figure 1 and Figure 6.

PCB 1 and PCB 2 name tags are not compatible in Figure 1.

Thank you very much for your comments. In the revised version, we tried to improve the quality of the figures.

Is the copper layer empty box or filled block? What is the reason or effect of this copper layer being half-sized? Is it change the isolation?

The proposed structure consists of two laminated PCBs with a copper layer between them. The copper layer contains a hole through which the SMA's dielectric passes. The copper layer serves as a shield for the transition. Due to the soldering between ports 1 and 2, there is no copper layer between ports 1 and 2. Copper is used here as a shield for transitions and as a spacer for accurate layer stacking.

My main concern is with the novelty of the design. The proposed main design topology is previously presented in [1]. In terms of performance, results could be sufficient. However, can the authors please address the uniqueness of the design in light of existing diplexing power dividers or similar design by comparing it with the previously published works? 

[1] doi: 10.1109/AP-S/USNC-URSI47032.2022.9887341.

Reference 1 presents the primary results of the initial design based on simulation only. This article presents the results of our work with in-depth details on the final enhanced structure and parametric studies for expanding the structure. Then, this final structure is fabricated based on the results of the analytical and parametric studies. In the article, the fabrication technique is clearly explained, and measurement results are provided. To compare the proposed work with similar works, a comparison table is provided in the revised version to express the novelties.

Reviewer 3 Report

The authors present a design of power divider with simple structures. The diplexing performance is clearly demonstrated for Ku band operation. Some suggestions/comments are:

(1) The introduction is too lengthy. I suggest to re-organize them to give a clear description on the motivation and novelty of the work.

(2) The figures need to be improved, especially Fig.3 and Fig.10.  Many words are not necessary, like "line-line", "black color".... 

(3) It may need more clearer description on the key factors that affect the range of the two bands and their bandwidth.   

(4) Again Fig.10, why stair type measurement results?

Structures and results are clearly explained.

Author Response

The authors present a design of power divider with simple structures. The diplexing performance is clearly demonstrated for Ku band operation. Some suggestions/comments are:

(1) The introduction is too lengthy. I suggest to re-organize them to give a clear description on the motivation and novelty of the work.

Your comments are greatly appreciated. We have carefully addressed the reviewers' comments in the revised version, and we have indicated the changes in red.

(2) The figures need to be improved, especially Fig.3 and Fig.10.  Many words are not necessary, like "line-line", "black color".... 

Chart quality is now improved by saving the pdf file from the word template. Since each figure contains several parameters and charts, the descriptions are provided to help the reader distinguish between the charts and parameters.

(3) It may need more clearer description on the key factors that affect the range of the two bands and their bandwidth.   

More explanations are added to the revised version. All modifications are indicated in the red color.

(4) Again Fig.10, why stair type measurement results?

During the measurement process, the obtained results had not been smoothed.

Reviewer 4 Report

The paper is well written and organized. It is recommended to accept after the following two minor revisions.

1. The proposed work Vs existing works in the open literature needed to be compared and tabulated with regards to the performance parameters.

2. Yes the proposed power divider works at Two bands in Ku, but there is no disgnated application is mentioned at those frequencies. I think application of satellite subsystems is too generic.

3. The measured Vs Simulated S parameter diagrams need to be plotted together and compared, and the resolution of measured diagrams need to be improved.  

4. It is advised authors to include a table of comparison between Simulated Vs measured performance parameters.

Minor english language errors need to be corrected and I advice authors to proofread the manuscript. 

Author Response

The paper is well written and organized. It is recommended to accept after the following two minor revisions.

  1. The proposed work Vs existing works in the open literature needed to be compared and tabulated with regards to the performance parameters.
  2. Yes the proposed power divider works at Two bands in Ku, but there is no disgnated application is mentioned at those frequencies. I think application of satellite subsystems is too generic.
  3. The measured Vs Simulated S parameter diagrams need to be plotted together and compared, and the resolution of measured diagrams need to be improved.  
  4. It is advised authors to include a table of comparison between Simulated Vs measured performance parameters.

Your comments are greatly appreciated. A comparison table is provided in the revised version to compare the proposed work with previous literature. In the revised version, we tried to improve the quality of figures. Since, there are many parameters in simulation and measurement, plotting them in a same figure could confuse the readers. So, the simulation results and measurement results are presented in a separate figure. A good agreement can be seen between the simulation and measured results.

About application of the proposed work and its relation to Sensors:

A Ku-band frequency band power divider is used as a subsystem of a satellite for the Earth observation and climate monitoring. This system can be equipped with various sensors and instruments to collect data on temperature, humidity, atmospheric composition, and other climate-related parameters from space satellites. The assigned frequency bands split the incoming and outgoing signals which is a commercial and critical feature for satellite subsystems. The split channel enables signals from the power divider onboard to be transmitted to processing units. These units process the data in real-time to extract meaningful information, such as temperature trends, cloud cover, and greenhouse gas concentrations. Therefore, the proposed Ku frequency power divider is crucial in ensuring efficient and reliable communication between satellite and ground stations. It allows the timely transmission of critical climate data. This is essential for monitoring and remote sensing climate change, understanding its impact, and formulating policies to address environmental challenges.

Round 2

Reviewer 1 Report

the authors have addressed most of my concerns

Minor editing of English language required

Author Response

Your review is greatly appreciated. In the revised version, we tried to edit paper grammatically. All modifications are indicated in red color.

Reviewer 2 Report

Necessary revisions have been made. 

It could be better to add one more reference to Table 1 which must be the same structure technology (SIW).

Author Response

Thank you very much for your comments. A similar work based on SIW technology has been added to table 1. In the revised version, red color indicates the modifications.

Reviewer 3 Report

It Ok for me  now.

Author Response

Thank you very much for your review